# A Numerical and Experimental Study on the Solidification Structure of Fe–Cr–Ni Steel Slab Casting by Roller Electromagnetic Stirring

**Hong Xiao** [1,2], **Pu Wang** [1,*], **Bing Yi** [2], **Xiqing Chen** [1], **Aiwu Li** [2], **Haiyan Tang** [1], **Weihong Li** [2] **and Jiaquan Zhang** [1,*]

[1] School of Metallurgical and Ecological Engineering, University of Science and Technology Beijing, Beijing 100083, China; xiaoh@cseco.cn (H.X.); s20180208@xs.ustb.edu.cn (X.C.); tanghaiyan@metall.ustb.edu.cn (H.T.)

[2] Electromagnetic Center, Hunan Zhongke Electric Co., Ltd., Yueyang 414000, China; yib@cseco.cn (B.Y.); liaw@cseco.cn (A.L.); liwh@cseco.cn (W.L.)

\* Correspondence: b20180114@xs.ustb.edu.cn (P.W.); jqzhang@metall.ustb.edu.cn (J.Z.); Tel.: +8618811139230 (P.W.); +8613911171237 (J.Z.)

**Abstract:** We present a segmented coupling model for slab casting by roller electromagnetic stirring (R-EMS) of electromagnetic, flow, heat transfer, and solidification behavior based on magnetohydrodynamics and solidification theory. A three-dimensional (3-D) segmented coupling model that included electromagnetic, flow, and heat transfer elements was established using Ansoft Maxwell and ANSYS Fluent software. The effects of the roller sleeve, magnetic shielding ring, coil, core, molten steel, and air domain on the electromagnetic, thermal and flow fields were studied numerically. The accuracy of the model was verified by measuring the magnetic flux density at the centerline in a pair of rollers and the electromagnetic force of the copper plate. Based on the numerical results of the optimal technical parameters, the effect of the R-EMS on the solidification of Fe–17 wt% Cr–0.6 wt% Ni stainless steel was explored. The results indicated that with each additional pair of electromagnetic rollers, the average electromagnetic force increased by 2969 N/m$^3$ in the casting direction, and 5600 N/m$^3$ in the central section of the rollers. With increasing number of pairs of rollers, the effective stirring region increased, and the velocity of molten steel at the solidification front first increased but then decreased. The strong electromagnetic swirling washing effect reduced the solidification rate of the slab shell and promoted the superheated dissipation of molten steel in the center of the strand. The center equiaxed crystal ratio of the slab was improved to 69% with two pairs of R-EMS rollers and electromagnetic parameters of 400 A/7 Hz, which was beneficial for obtaining a uniform and dense solidified structure to improve the subsequent hot working performance and product quality.

**Keywords:** roller electromagnetic stirring (R-EMS); Fe–17 wt% Cr–0.6 wt% Ni steel; solidification structure; number of pairs for rollers; washing effect

## 1. Introduction

Nickel-saving stainless steel alloys, such as Fe–17 wt% Cr–0.6wt% Ni, have a ferrite structure at room and high temperatures that causes them to develop columnar crystals easily during solidification with a very low ratio of equiaxed crystal in steel castings. The columnar crystal structure has obvious anisotropy in the subsequent plastic working process, which is likely to produce wrinkle-like defects in plate products due to poor deep drawability. Controlling the cast structure and obtaining a high ratio of equiaxed crystals during the continuous casting (CC) process is important to improve the product's subsequent hot working performance and quality; this has always been a central issue for the steel industry [1,2]. Kunstreich et al. [3] addressed the surface/subsurface quality (slivers, pipe, pinholes, blowholes, inclusion content) and diversion rate (slab casting abnormality

codes) of products cast on high or low throughput thick slab machines. They found that slow speed or wide slab machines that create or maintain a stable double roll flow in the mold are key to eliminating slab defects, although the intensity of the double roll flow pattern must not be excessive. Electromagnetic stirring (EMS) drives the flow of molten steel to control heat and mass transfer behavior through an inductive non-contact electromagnetic force (EMF). The arrangement and use of EMS to control metallurgical behavior in the secondary cooling zone (SCZ) to improve the quality of strands has received less research attention.

It has been shown that the transfer behavior of molten steel in the SCZ during slab casting directly affects the internal quality of the strand and can simultaneously control the quality of the rolled materials by adjusting the flow of molten steel and temperature [4]. The SCZ of the slab is mainly equipped with insertion-type EMS (Nippon Steel, Tokyo, Japan), box-type EMS (ABB, Zurich, Switzerland), and roller electromagnetic stirring (R-EMS) (Danieli Rotelec, Paris, France) [5]. Compared with insertion-type and box-type EMS, roller electromagnetic stirring (R-EMS) has the coil inside the roller and replaces the support roller for the strand, and has a higher EMF to push the internal molten steel flow. Lei et al. [6,7] investigated the magnetic field and flow field distribution of three R-EMS modes (double-disc, double-ring, and triple-ring). EMS in the double-ring mode was the most efficient, producing a larger area of circulation flow inside the strand at the same power as the other modes. With increasing frequency, the magnetic flux density at the center of the slab decreased and the EMF and velocity of the molten steel increased, indicating that the EMF is a direct indicator of the effect of R-EMS. Shen et al. [8] established a coupled model for the flow and solidification behavior of the slab in SCZ based on Maxwell's equations and the k-epsilon model, and they observed that the molten steel flow direction was consistent with the EMF direction and that the stirring effect was significantly weakened with increasing thickness of the solidified shell. Wang et al. [9] suggested that the EMF generated by the traveling wave magnetic field was concentrated near the wide surface of the strand and that the EMF produces a horizontal stirring flow. Jiang et al. [10] established a three-dimensional macroscopic transport model for the slab and observed that the endpoint of solidification was at the one-quarter position in the direction of the width of the slab. Compared with a linear stirrer pushed to one side, a rotary stirrer in the SCZ favors uniform solute distribution at the solidification end. Wang et al. [11] analyzed the factors contributing to the low equiaxed crystal ratio of 430 ferrite stainless steel. When there is insufficient EMS, the equiaxed crystal ratio is greatly improved by adjusting the SCZ and casting speed, which reduces the cross-cracking of the strand. Zhou et al. [12] analyzed the mechanism of the effect of EMS on the solidification structure of martensitic stainless steel in slab casting and observed that the central equiaxed crystal ratio of the strand reached an average of 50% and up to 57% using the appropriate magnetic flux density. Research by Nippon Steel [13] has shown that R-EMS can control the effective flow of internal molten steel to reduce the columnar crystal structure of electrical steel and stainless steel, simultaneously increasing the equiaxed crystal ratio to improve central shrinkage, porosity, and segregation of the slab, which is conducive to improving the casting speed and quality of the slab production process.

These studies indicate that R-EMS can replace the normal pinch roller in the CC machine with different pairs of rollers. Different arrangements of the rollers in the segments will have varied magnetic field distributions, areas of EMF, and molten steel flow patterns. Considering the importance of metallurgical behavior in the SCZ of a slab casting with regard to controlling the internal quality of the strand, a coupled model of three-dimensional (3-D) electromagnetic, flow, heat transfer, and solidification behavior in the SCZ has been developed in this study. We used Fe–17 wt% Cr–0.6 wt% Ni steel to investigate the effects of different numbers of pairs of rollers for R-EMS on the magnetic field distribution and solidification behavior. We aimed to use numerical modeling to provide theoretical guidance for the improvement of the solidification structure and internal quality of ferrite stainless steel castings.

## 2. Methods

### 2.1. Numerical Model Description

The structure of EMS equipment mainly consists of a roller sleeve (copper), a magnetic shielding ring, a coil, a core, molten steel, and an air domain (Figure 1; air domain not shown). The magnetic shielding ring consists of a section of a ring with the remainder filled with air. The thermophysical parameters and CC process parameters used in the simulation calculations are given in Table 1. The origin of coordinates in the model is at the center of the mold meniscus, where the casting direction is along the positive Z-axis, while the X- and Y-axes are parallel to the narrow and wide sides of the strand, respectively. The computational domain model was developed with R-EMS in the SCZ for the production of Fe–17 wt% Cr–0.6 wt% Ni steel with a cross-section of 1280 mm × 200 mm. The R-EMS structure is linear, with five coils winding around the roller with a diameter of 240 mm and length of 1550 mm. The three pairs of rollers were 4.159, 3.911, and 3.660 m away from the meniscus, and the continuous linear stirring mode was used.

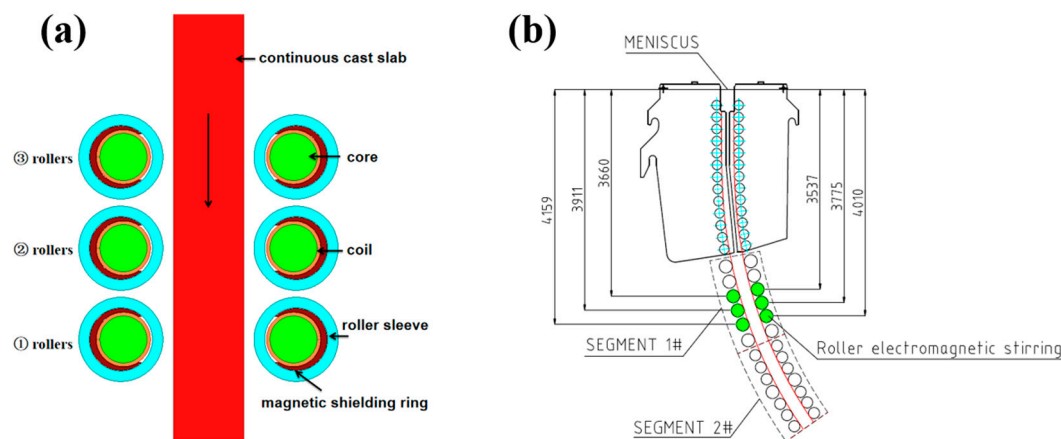

**Figure 1.** Roller electromagnetic stirring (R-EMS): (**a**) Structure diagram; (**b**) installation location diagram (units: mm).

**Table 1.** Geometrical and operational parameters of the cast and thermophysical properties of Fe–17 wt% Cr–0.6 wt% Ni steel.

| Parameter | Value |
|---|---|
| Sectional dimension (mm$^2$) | 1280 × 200 |
| Stirring position (m) | 4.159, 3.911, 3.66 |
| Phase number | 2 |
| Casting speed (m·min$^{-1}$) | 0.9 |
| Specific water flow (L·kg$^{-1}$) | 0.4 |
| Relative permeability of each material | 1 |
| Relative permeability of the iron core | 1000 |
| Electrical conductivity of steel (S·m$^{-1}$) | 7.14 × 10$^5$ |
| Electrical conductivity of copper (S·m$^{-1}$) | 3.18 × 10$^7$ |
| Thermal conductivity (m·K$^{-1}$) | 29 |
| Steel density (kg·m$^{-3}$) | 7000 |
| Steel viscosity (m·s$^{-1}$) | 0.006 |
| Liquids temperature (K) | 1768 |
| Solids temperature (K) | 1703 |
| Specific heat (kg·K$^{-1}$) | 690 |
| Latent heat (J·kg$^{-1}$) | 275,000 |
| Superheat degree (K) | 30 |

Due to the magnetic Reynolds number $R_m < 1$ during electromagnetic stirring in the CC process, the effect of steel flow on the external magnetic field was negligible. The effect of the solid and liquid phases of steel with little different electrical conductivity was ignored in the high-temperature zone on the electromagnetic force. The development of coupled equations for the electromagnetic field, flow field, heat transfer, and solidification behavior are relatively mature, as described in detail by Li et al. [14] and Wang et al. [15]. The principles of the linear stirrer are shown in Figure 2 [16]. The electromagnetic roller is a traveling wave magnetic field stirrer, meaning the iron core and the magnetic circuit are disconnected and that the electromagnetic thrust toward one side controls the linear motion of the molten steel.

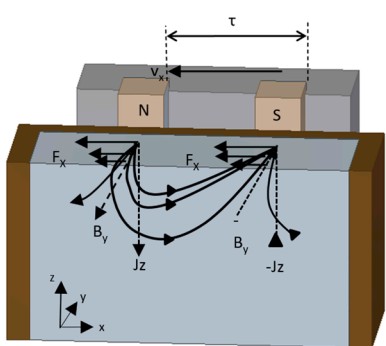

**Figure 2.** Working principle of the linear stirrer.

### 2.2. Boundary Conditions and Numerical Solution Procedure

For the electromagnetic field, a tetrahedral mesh with a mesh number of 518,230 was used in the electromagnetic model. For the pairs of rollers, each R-EMS had five coils loaded with two-phase alternating current, and the phase difference of each phase was $90°$. The magnetic line was parallel to the surface of the air unit that surrounded the stirrer. The insulation boundary conditions were set between the coil, copper tube, and iron core.

For the calculation of flow and solidification, a segmented model was established with no electromagnetic force in the mold and a foot-roll region to calculate the solidification and flow information; a hexahedral mesh was used for the fluid calculation. Grids were refined in areas with intense transmission density, such as the nozzle boundary layer and the solidification region, resulting in a total of approximately 3 million grids. The residual values for energy were smaller than $10^{-6}$ and others were smaller than $10^{-4}$. ANSYS Fluent 16.0 (ANSYS, Inc., Canonsburg, PA, USA) was used to judge the convergence during the calculation. The "Profile" module in ANSYS Fluent was used to extract the first segment of the computational domain's exit data as an entry condition for the second segment. To ensure the effective range of EMF action and the full development of turbulence flow, 3–4.8 m of SCZ was selected in this study for the computational domain. The ANSOFT Maxwell (ANSYS, Inc., Canonsburg, PA, USA) simulation was used to obtain the electromagnetic field data of the computational domain, and Fluent software was used to calculate the flow, heat transfer, and solidification information with the steady state in the SCZ. The node coordinate information in Fluent was loaded into Maxwell, and the time-averaged EMF was extracted using the coordinate interpolation algorithm. Finally, the EMF was loaded into the momentum equation using the User Defined Function (UDF). The model boundary conditions were as follows:

1. Computational domain inlet: The velocity and temperature of the first computational domain exit and the liquid phase fraction information were loaded as inlet boundary conditions.
2. Calculating domain outlet: Zero gradients for all physical quantities in the direction of the export normal using fully developed boundary conditions.

3.  Wall: The cooling conditions were described using the convective heat transfer coefficient [10].

### 2.3. Experimental Procedure

The magnetic flux density was measured using a LakeShore Digital Signal Processing Mode Tesla Meter 475 (Zhongke Electric, Hunan, China). The EMF was tested using a thrustmeter manufactured in-house, as shown in Figure 3. The principle of the testing device was based on the measurement method of the copper plate simulating the strand. A number of thin copper plates with 2 mm thickness were evenly distributed and suspended symmetrically and in parallel with the thickness of the strand between the working surfaces of the EMS. The electromagnetic thrust received on each copper plate was measured separately with a tension transducer, with each copper plate representing the thrust received by a strand with a certain thickness in the corresponding position.

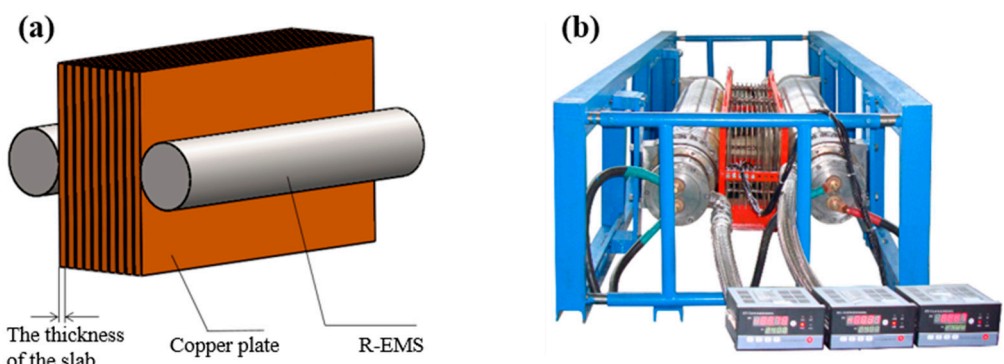

**Figure 3.** Copperplate measurement method: (**a**) Structure diagram; (**b**) physical device diagram.

The main chemical components of the Fe–17 wt% Cr–0.6 wt% Ni steel are listed in Table 2. The sampling location of the slab cross-section produced under the corresponding working conditions is shown in Figure 4. The cross-section of the sample under each working condition was flattened using a lathe and polished using a milling machine, such that there were no processing marks affecting the observation of the inspection surface. An aqueous solution of industrial hydrochloric acid at a volumetric ratio of 1:1 was used as an erosion agent. The surface-finished samples were immersed in the acid etchant and eroded at a water bath temperature of 70 °C for 20 min. Immediately after erosion, the surface was rinsed with water and blown dry with a high-pressure airflow, images were obtained, and the equiaxed crystal ratio was recorded by Image-Pro Plus (Media cybernetics, Inc., Rockville, MD, USA).

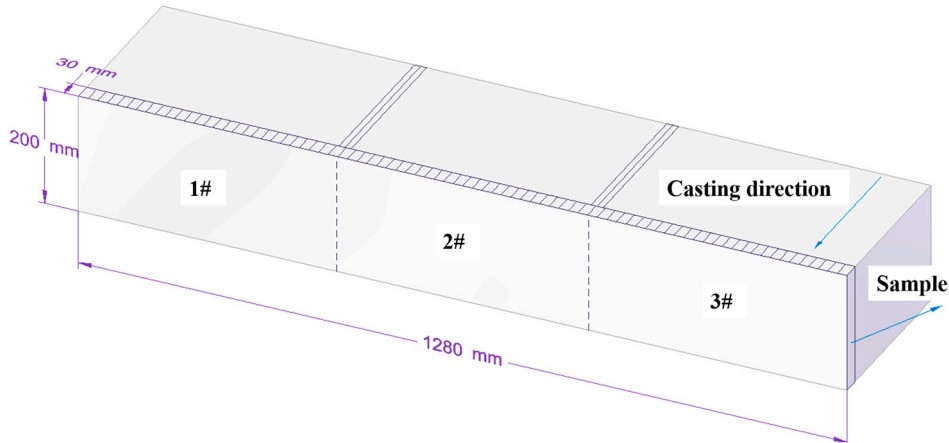

**Figure 4.** Sample cutting from slab casting for metallographic analysis.

**Table 2.** Chemical composition of the Fe–17 wt% Cr–0.6 wt% Ni steel used in this study (units: wt%).

| C | Cr | Ni | Mn | P | S | Si |
|---|---|---|---|---|---|---|
| ≤0.12 | 17 | 0.6 | ≤1.25 | ≤0.035 | ≤0.03 | ≤0.75 |

## 3. Results and Discussion

### 3.1. Analysis of Electromagnetic Field

Figure 5a shows a comparison of the calculated and measured values of magnetic flux density, and Figure 5b shows the EMF on the centerline of the wide surface with a pair of rollers. The measured and calculated values of magnetic flux density on the centerline of the roller and the EMF of the copper plate are roughly in agreement, which verifies the reliability of the model to an acceptable level. Figure 5b shows that the EMF increased rapidly and then decreased slowly as the frequency increased, and the greatest EMF of a pair of roller stirrers was obtained at a frequency of 9 Hz.

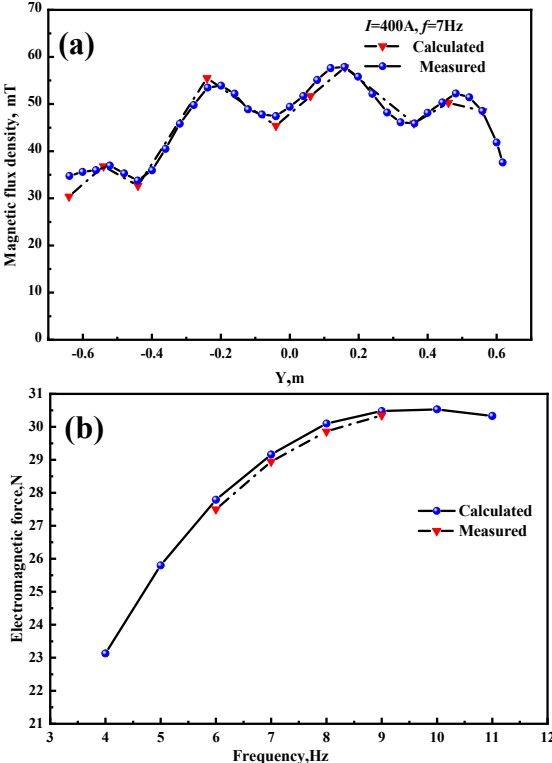

**Figure 5.** Comparison of measured and calculated values on the centerline of the wide face of the pair of rollers: (**a**) magnetic flux density, (**b**) electromagnetic force.

Figure 6a–c presents the distribution of the magnetic flux density on the surface of the slab for a current intensity of 400 A and a frequency of 7 Hz under cases of one pair, two pairs and three pairs, respectively, in which the effective area of the magnetic flux density increased with an increasing number of rollers. The traveling wave magnetic field had a certain directionality which produced an end effect, resulting in a larger magnetic flux density on the thrust side (right side of the strand in Figure 6) than on the start side (left side of the strand in Figure 6).

Figure 7a reveals the distribution of the EMF along the centerline in the casting direction under a different number of pairs of rollers for a current of 400 A and frequency of 7 Hz, and Figure 7b shows the distribution of the EMF along the centerline of the rollers in the wide direction. For one, two, and three pairs of rollers, the maximum EMF on the centerline of the slab along the casting direction was 12,090, 18,573, and 21,229 N/m$^3$,

respectively, and the average EMF was 2023, 5066, and 7962 N/m$^3$, respectively. The maximum EMF on the centerline of the wide surface for each pair of rollers was 12,354, 18,084, and 22,874 N/m$^3$, respectively, and the average EMF was 10,247, 15,730, and 21,336 N/m$^3$, respectively. The maximum force was located on the thrust side of the slab, and the EMF of the molten steel increased with increasing number of pairs of rollers.

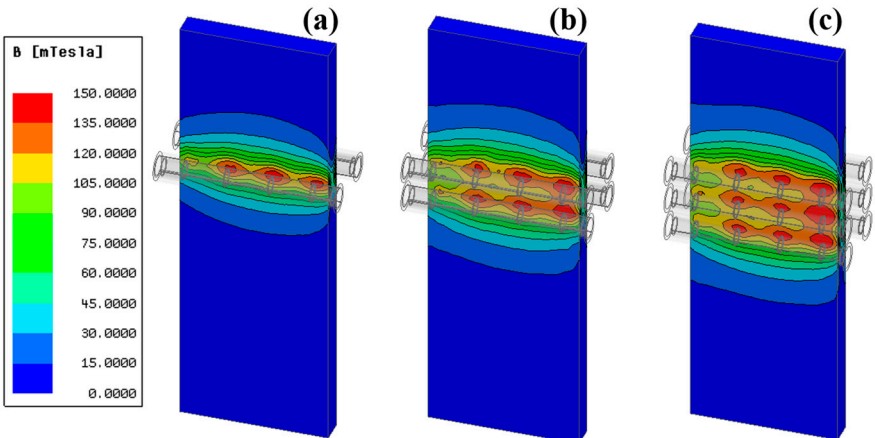

**Figure 6.** Contour of magnetic flux density on the surface of the stand with (**a**) one pair, (**b**) two pairs, and (**c**) three pairs of rollers.

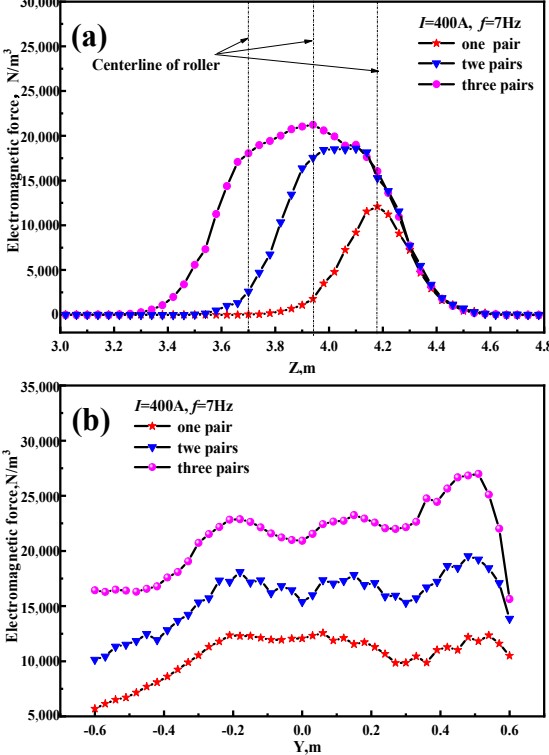

**Figure 7.** Distribution of internal electromagnetic force in the strand with different number of rollers (**a**) along the centerline in the casting direction, (**b**) along the centerline of the rollers in the wide direction.

Figure 8a shows the distribution of the EMF in the casting direction under two pairs of rollers at different frequencies, and Figure 8b reveals the distribution of the EMF in the casting direction under the two pairs of rollers at different currents. The distribution of the EMF indicates that it was small at both ends, large in the middle, and evenly

distributed between the rollers. The maximum EMF at the center of the strand increased from 4750 to 19,000 N/m$^3$ as the current intensity increased from 200 to 400 A. In addition, the maximum EMF at the center of the strand decreased from 20,838 to 17,995 N/m$^3$ when the frequency increased from 4 to 8 Hz. The strand exhibited a certain magnetic conductivity when the magnetic induction lines from the air into the strand deviated, clustering in one location and forming a magnetic shield. The difference in magnetic flux between the interior and edges of the strand resulted in an uneven distribution of the induced current, which was mostly concentrated on the surface of the slab, a phenomenon known as the "skin effect". This effect leads to a reduction in the penetration of the magnetic field at higher frequencies [17]. It shows that the solidified shell with a certain electrical conductivity has a certain shielding effect on the magnetic field, and therefore the central magnetic induction intensity decreases slightly as the current frequency increases.

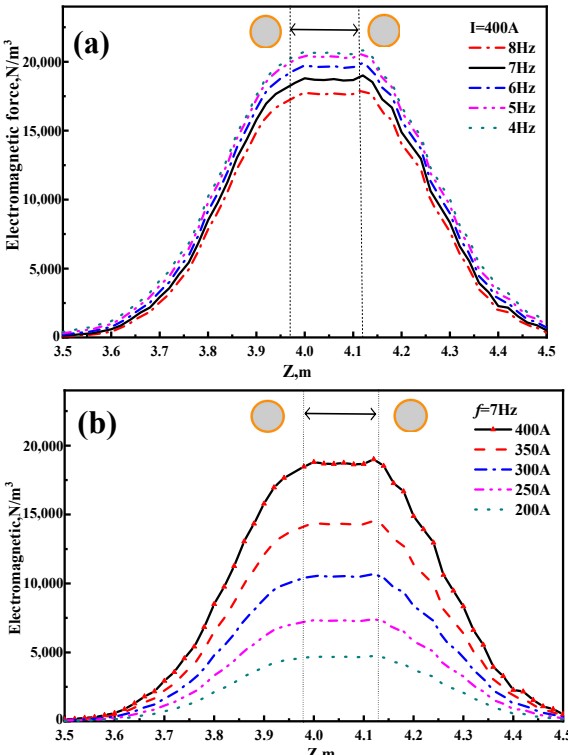

**Figure 8.** Distribution of the electromagnetic force in the casting direction under the two pairs of rollers at (**a**) different frequencies and under (**b**) different currents.

### 3.2. Analysis of Flow and Solidification Behavior

Figure 9a demonstrates the velocity distribution along the centerline in the casting direction on the characteristic line of molten steel with a different number of pairs of rollers, and Figure 9b shows the velocity distribution along the centerline of the rollers in the wide direction. An increase in the number of rollers led to an increase in the local volume of the EMF on the strand, and the EMF was the driving force of the flow of molten steel to wash the solidification front in the SCZ. The effective washing velocity range—defined as the range over which the flow velocity is greater than the casting speed—of the solidification front along the casting direction was 4.0–4.35 m, 3.8–4.35 m, and 3.6–4.35 m for one, two, and three pairs of rollers, respectively, and the maximum washing velocity was 0.7, 0.8, and 0.76 m/s, respectively. Zhang et al. [18] found that the high-velocity jet flow from the side holes can lead a larger turbulence zone in the mold zone and part of the SCZ. Although the EMF of two pairs of rollers is lower than that of the three pairs, the washing region of the two pairs is further down, leaving lower turbulent kinetic energy intensity in the offset mold area. Thus, the strand has a greater maximum washing velocity with two pairs

of rollers than with three pairs. Figure 8b shows that the maximum flow velocity under different numbers of pairs of rollers was distributed on one side of the strand. The flow velocity on the thrust side of the EMF was greater than on the start side, which is roughly in agreement with the motion characteristics of the traveling wave magnetic field.

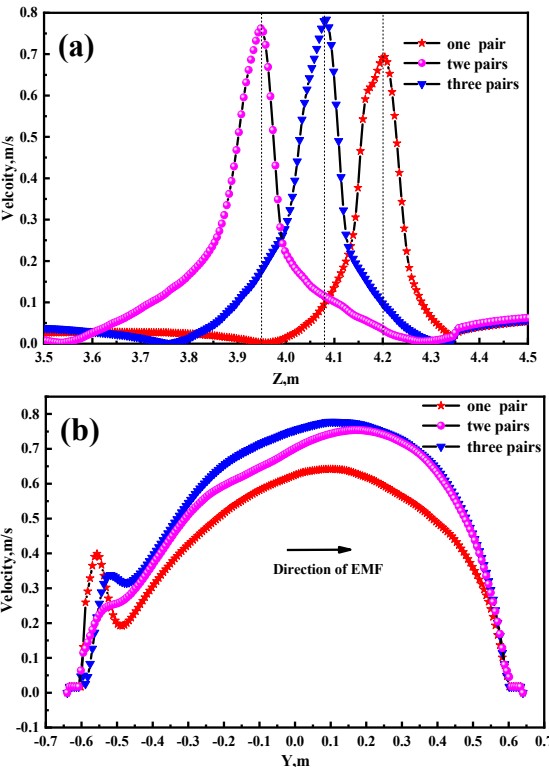

**Figure 9.** Distribution of velocity with different numbers of rollers (**a**) along the centerline in the casting direction, and (**b**) along the centerline of the rollers in the wide direction.

Figure 10a–d shows the temperature distribution and molten steel streamline on the center surface of the narrow face in the slab with 0–3 pairs of rollers. The EMF caused the molten steel to move from one side of the narrow surface to the other, and the continuity of the flow to the narrow solidification front led to the formation of an upper and lower circulation of the molten steel, resulting in a uniform core temperature and mixing of the slab. With an increasing number of pairs of rollers, the area of molten steel flow at the cross-section expanded and the forced heat exchange between the central high-temperature molten steel and solidified shell led to a larger low-temperature zone in the center of the strand. According to solidification theory, a lower temperature of the central molten steel is more conducive to the formation of nucleation particles. Xu et al. pointed out [19] that the washing of molten steel against the solidification front may cause the "melting" of the dendrite arm to provide nucleation particles for the formation of equiaxed crystals, which ultimately increases the central equiaxed crystal ratio of the strand.

Figure 11a illustrates the variation of the solidified shell at the start side along the casting direction at the center of the narrow face for the strand with different numbers of pairs of rollers, and Figure 11b shows the change in shell thickness at the thrust side along the casting direction at the center of the narrow face for the strand with different numbers of pairs of rollers. The solidification front is considered the location where the liquid phase fraction is 0.3. For zero, one, two, and three pairs of rollers, the thickness of the solidified shell at the outlet of the calculation domain was 42.37, 40.96, 40.14, and 38.43 mm on the start side of the EMF, respectively, and 42.37, 42.27, 37.62, and 37.60 mm on the thrust side of the EMF, respectively. The high-velocity flow of molten steel rushes to the solidification front and interrupts some of the columnar crystals, resulting in the

slow growth of the solidified shell in the stirring region. The solidification rate on the electromagnetic thrust side was significantly lower than the start side, which roughly coincides with the characteristics of the traveling wave magnetic field.

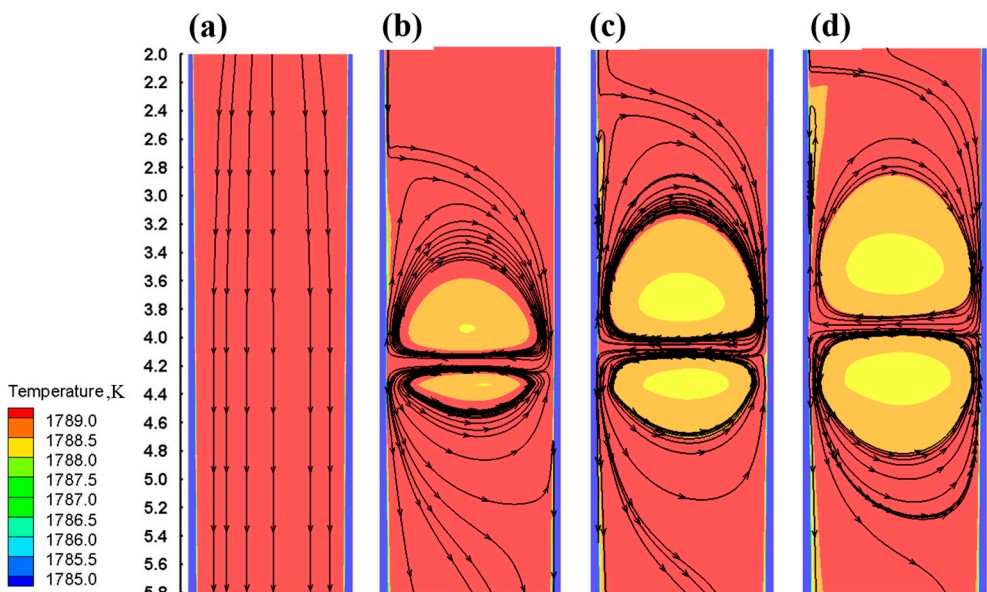

**Figure 10.** Temperature distribution and flow on the narrow center surface of the strand with (**a**) zero pairs, (**b**) one pair, (**c**) two pairs, and (**d**) three pairs of rollers.

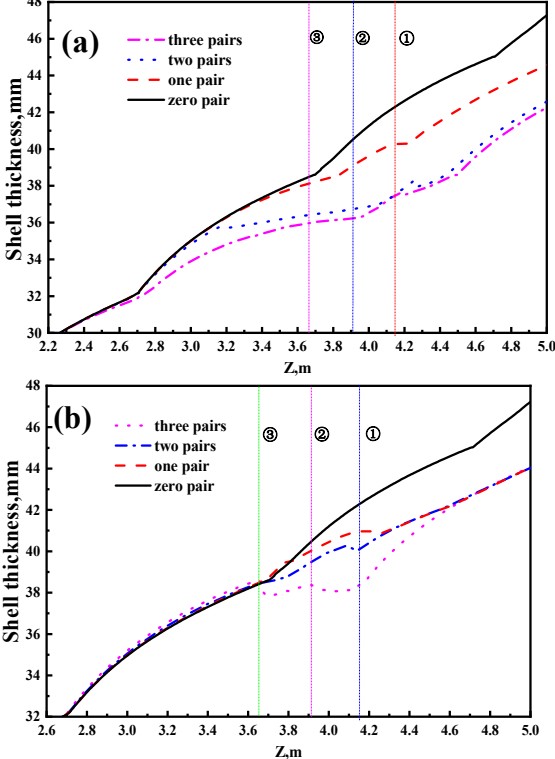

**Figure 11.** Distribution of the shell thickness at the narrow center face of the strand on the (**a**) start side and (**b**) thrust side.

### 3.3. Experiments of Solidification Structure Obtained by R-EMS

Two pairs of rollers were selected for Fe–17 wt% Cr–0.6 wt% Ni steel slab casting in the experiments of solidification structure control by R-EMS. The semisolid zone in the center of the slab when two pairs of rollers were used was larger than when one pair of rollers was used. Although the EMF was smaller than when three pairs were used, the washing velocity of the solidification front was greater with two pairs than three pairs, which was beneficial for the formation of equiaxed crystals in the strand. In addition, the cost of instrumentation and power consumption are lower when two pairs of rollers are used. The solidification structures of the slab produced when the R-EMS was turned off and on were compared during the experiment, as shown in Figure 12. When the R-EMS was turned off, the macrostructure of the slab was more developed in the columnar crystal, which is related to the characteristics of Fe–17 wt% Cr–0.6 wt% Ni steel. Having a content of Cr in the steel greater than 16% led to a solidification process without the $\alpha \rightarrow \gamma$ phase transition process, with the ferrite structure maintained. Pang et al. [20] found that there was no phase transition to hinder the development of columnar crystals during the process of grain growth; thus, the grain size was coarse, and the chemical elements were prone to segregation, which can seriously affect product quality. When the R-EMS was turned on with electromagnetic parameters of 400 A and 7 Hz, the EMF generated by the traveling wave magnetic field caused the molten steel to flow violently and wash the columnar crystal front to reduce the temperature gradient at the solidification front, inhibiting the growth of columnar crystals. At the same time, the high-velocity flow of the molten steel can break up the columnar dendrite arm to form free nuclei in the central low-temperature area. Finally, the central equiaxed crystal ratio of the strand was increased to 69%.

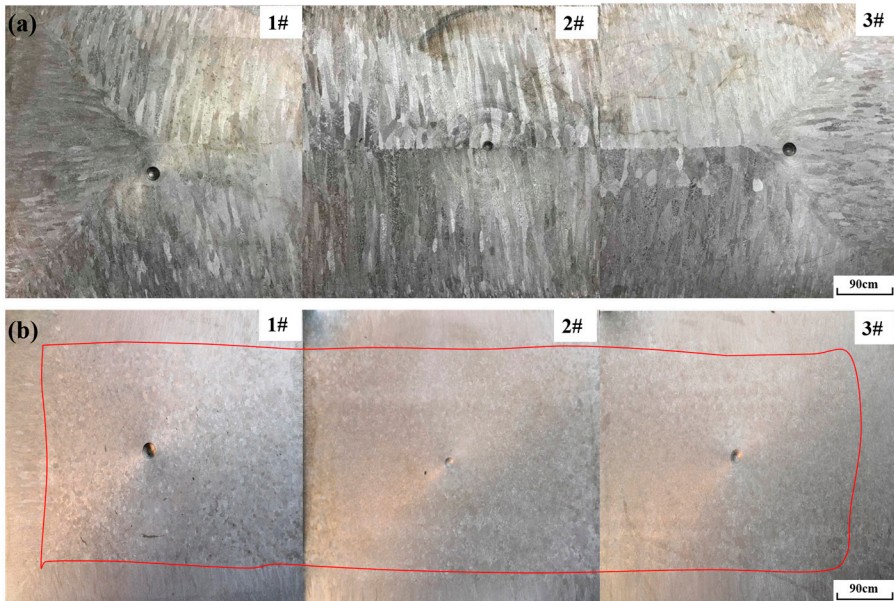

**Figure 12.** Cross-section of the as-cast macrostructure of the strand (**a**) without R-EMS and (**b**) with two pairs of rollers used in R-EMS (at 400 A and 7 Hz).

## 4. Conclusions

Here, a 3-D segmented coupling model for electromagnetic, flow, and heat transfer behavior was established for the slab casting of stainless steel. The effects of R-EMS on the magnetic field distribution and solidification behavior were revealed, and the optimal technical parameters to control the as-cast macrostructure of steel Fe–17 wt% Cr–0.6 wt% Ni have been presented. The main conclusions are as follows:

1.  The characteristics of the traveling wave magnetic field of the R-EMS in the SCZ will produce a maximum EMF being located on the starting side of the slab strand.

For each additional pair of electromagnetic rollers, the average EMF in the casting direction increases by 2969 N/m$^3$, and the average EMF in the center section of the rollers increases by 5600 N/m$^3$.

2. With an increasing number of pairs of stirring rollers, the effective stirring area of the molten steel inside the strand is enlarged by the EMF, and the velocity of molten steel at the solidification front first increases and then decreases. The flow washing effect from the strong electromagnetic force will reduce the solidification rate of the local shell and accelerate the superheated dissipation of the molten steel center, which is beneficial for the formation of equiaxed crystal.

3. The use of two pairs of electromagnetic rollers at 400 A and 7 Hz can produce a center equiaxed crystal ratio of 69% in the slab strand of 200 mm $\times$ 1280 mm, which helps to improve its hot working behavior.

**Author Contributions:** Conceptualization, H.X. and B.Y.; methodology, H.X. and P.W.; investigation, B.Y. and X.C.; resources, A.L. and W.L.; writing—original draft preparation, H.X. and P.W.; writing—review and editing, H.X., P.W. and J.Z.; visualization, X.C. and P.W.; supervision, A.L., H.T. and J.Z.; project administration, H.T. and J.Z.; H.X. and P.W. are co-first authors. All authors have read and agreed to the published version of the manuscript.

**Funding:** This research was funded by the Beijing Municipal Natural Science Foundation (BJNSF) (Grant No.2182038) and National Natural Science Foundation of China (NSFC) (Grant No.51874033 and No.U1860111), National Key R&D Program of China (Grand No.016YEB0601302).

**Acknowledgments:** The authors give thanks for the industrial test in Hunan Valin Lianyuan Iron & Steel Croup Co., Ltd.

**Conflicts of Interest:** The authors declare no conflict of interest.

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
