# Peer review of "A Numerical and Experimental Study on the Solidification Structure of Fe–Cr–Ni Steel Slab Casting by Roller Electromagnetic Stirring"

_metals, doi:10.3390/met11010006_

Round 1
Reviewer 1 Report
1. In abstract, the first sentence mentions "magnetic fluid..". "Magnetic fluid" is one of the well-known names for ferrofluid media (https://en.wikipedia.org/wiki/Ferrofluid). It is recommended to replace "... based on magnetic fluid and solidification theory" with "based on magnetic, fluid and solidification theory" or "... based on magnetohydrodynamics and solidification theory" (better).
2. Please describe clearly and carefully the aim of the work in the Introduction.
3. Describe the shape of the "magnetic shielding ring". Figure 1 shows that it has the shape of a ring sector. If so, describe what the remaining sector is filled with and the properties of this material.
4. Add the properties of the material (electrical conductivity, thermal conductivity) that the roller sleeve is made of.
5. The description of how the rotational stirrer works (Figure 2) is outside the scope of this article. It is recommended to remove.
6. In calculations with an alternating magnetic field, a very important characteristic is the value of the skin depth (https://en.wikipedia.org/wiki/Skin_effect). Add an estimate of the skin depth range for the range of the studied magnetic field frequencies. Add a comparison of the number of elements of the computational mesh across the ingot with the skin depth. It is necessary to estimate the reliability of the calculations.
7. Add also an estimation of the skin depth and mesh size for the modeling of the electromagnetic process in copper plates. Do the outer copper plates shield the magnetic field in the center plates?
8. During solidification, the ingot region becomes two-phase. The solid and liquid phases of steel have different electrical conductivity. This significantly affects the distribution of the electromagnetic force during the solidification process. Describe how this effect was taken into account in the calculations. If this effect was not taken into account, add an estimate of its effect on the resulting electromagnetic force.
9. On page 4 it is mentioned that the flow is turbulent. Add an estimate of the hydrodynamic Reynolds number for different calculation variants. Describe which turbulence model was used, the reason for choosing this model and the parameters of this model.
10. What the velocities are shown in Figure 9: average, instantaneous, or steady-state? In such processes, an intense turbulent flow is usually not stationary. Vortex structures migrate during the process. This is good. The most unstable flow usually leads to the best mixing. If the flow is not stationary in your study, please show the oscillation characteristic (rms(V)) on the velocity profiles (something like errorbars) or on an additional figure (rms(V) versus Z).
11. (Page 10, top) It is recommended to replace 'The solidification structures of the slab produced when the R-EMS was closed and opened were compared, as shown in Figure 12.' with 'The solidification structures of the slab produced when the R-EMS was turned off and turned on were compared, as shown in Figure 12.'
12. Describe how the center equiaxed crystal ratio of the slab (69%) was obtained.
13. In Figure 6 we see an asymmetric field of electromagnetic force relatively the vertical axis. Is it possible to recommend changing the direction of the running magnetic field for each roller as follows? The electromagnetic force of the neighboring rollers, and, as a consequence, the generated flow, must be directed opposite. This will create a less asymmetrical force and flow field. If so, I suggest adding it to the conclusions as a recommendation.
14. Possible missing a word in 'There was no phase transition to hinder the columnar crystals during the process of grain growth;' (page 10). Maybe 'There was no phase transition to hinder appearance/develop of the columnar crystals during the process of grain growth;' ?
15. Typo '2.3. Experimetal' (page 5).
16. Typo '..current density of 400 A..' - '..current intensity/amperage/strength..'? (page 6)
17. Typo '3. B C W A , B Y Y A , C J Y A B , et al....' (page 12).
18. Typo '5. SKunstreich. Electromagnetic ...' (page 12).
19. I recommend to improve English a bit.
Author Response
Dear reviewer and editor,
Thank you very much for your kind and responsible reviewing. The comments are of great importance to the improvement of our paper. After careful modifying and polishing, the revision has been finished. The response on the comments is uploaded in an attachment.
Best regards,
Pu Wang on behalf of all authors

Reviewer 2 Report
Dear authors,
thank you very much for your interesting manuscript.
Please, follow my suggestions to improve the final high scientific quality of your paper:
Line 15-17, Abstract: Please, control the English meaning of the sentence “ Experiments on the effect of the number of pairs 15 of rollers in a 1280 mm×200 mm section on the macrostructure of the continuous casting slab were 16 also conducted”.
Line 19-20: units “ N/m3”…there are missing the upper index.
Line 34, Introduction. Please, it would be great to add some appropriate references about the slab defects depending on casting boundary conditions.
Line 61, Introduction: Please, specify more precisely „ velocity of the molten steel”. I suppose that you mean "the steel flow velocity inside of solidified casting strand during EMS".
Line 92, title 2.1. Physical Model: Please, maybe it will be much better to use Numerical model setting/ or Numerical model description?
Line 104, Figure1: Please, enlarge the description in figures (the size of letters).
Line 108-109, Table 1: units! There are missing the upper indexes.
Line 149, Title 2.3. Please, use better title, for example “Experimental procedure”?
Line 170, Figure 4: In the figure, there is blue arrow without description.
Line 180, Figure 5. In legend, there is typing error “Claculated” instead “Calculated”.
The text for Figures 5, 6, 7, 8, 9, 10, 11: in the text to figures, please, specify more precisely the description which correspondents to 5(a) and to 5(b) for example. The same is for Figure 6 -11.
Figures 5 -11: Please, enlarge the description in figures (the size of letters).
Line 214, ref.17. It would be better to describe the ref. 17 depending on your achieved results. Probably, you wanted to say, that skin effect was also achieved in your previous work Ref.17. But it must be clear for readers. The same is with Ref 18 (Line 227), Ref. 19 (Line 248) and Ref. 20 (Line 277). Thank you.
Thank you very much for your cooperation.
Reviewer
Author Response

(The authors gave the same response as above.)
